# LEARNING TO COMMUNICATE THROUGH IMAGINATION WITH MODEL-BASED DEEP MULTI-AGENT REINFORCEMENT LEARNING

## ABSTRACT

The human imagination is an integral component of our intelligence. Furthermore, the core utility of our imagination is deeply coupled with communication. Language, argued to have been developed through complex interaction within growing collective societies serves as an instruction to the imagination, giving us the ability to share abstract mental representations and perform joint spatiotemporal planning. In this paper, we explore communication through imagination with multi-agent reinforcement learning. Specifically, we develop a model-based approach where agents jointly plan through recurrent communication of their respective predictions of the future. Each agent has access to a learned world model capable of producing model rollouts of future states and predicted rewards, conditioned on the actions sampled from the agent's policy. These rollouts are then encoded into messages and used to learn a communication protocol during training via differentiable message passing. We highlight the benefits of our model-based approach, compared to a set of strong baselines, by developing a set of specialised experiments using novel as well as well-known multi-agent environments.

## 1 INTRODUCTION

"We use imagination in our ordinary perception of the world. This perception cannot be separated from interpretation." (Warnock, 1976). The human brain, and the mind that emerges from its working, is currently our best example of a general purpose intelligent learning system. And our ability to imagine, is an integral part of it (Abraham, 2020). The imagination is furthermore intimately connected to other parts of our cognition such as our use of language (Shulman, 2012). In fact, Dor (2015) argues that:

> "The functional specificity of language lies in the very particular functional strategy it employs. It is dedicated to the *systematic instruction of imagination*: we use it to communicate directly with our interlocutors' imaginations."

However, the origin of language resides not only in individual cognition, but in society (Von Humboldt, 1999), grounded in part through interpersonal experience (Bisk et al., 2020). The complexity of the world necessitates our use of individual mental models (Forrester, 1971), to store abstract representations of the information we perceive through the direct experiences of our senses (Chang and Tsao, 2017). As society expanded, the sharing of direct experiences within groups reached its limit. Growing societies could only continue to function through the invention of language, a unique and effective communication protocol where a sender's coded message of abstract mental representations delivered through speech, could serve as a direct instruction to the receiver's imagination (Dor, 2015). Therefore, the combination of language and imagination gave us the ability to solve complex tasks by performing abstract reasoning (Perkins, 1985) and joint spatiotemporal planning (Reuland, 2010).

In this work, we explore a plausible learning system architecture for the development of an artificial multi-agent communication protocol of the imagination. Based on the above discussion, the minimum set of required features of such a system include: (1) that it be constructed from multiple individual agents where, (2) each agent possesses an abstract model of the world that can serve as an imagination, (3) has access to a communication medium, or channel, and (4) jointly learns and interacts in a

collective society. Consequently, these features map most directly onto the learning framework of model-based deep multi-agent reinforcement learning.

Reinforcement learning (RL) has demonstrated close connections with neuroscientific models of learning (Barto, 1995; Schultz et al., 1997). However, beside this connection, RL has proven to be an extremely useful computational framework for building effective artificial learning systems (Sutton and Barto, 2018). This is true, not only in simulated environments and games (Mnih et al., 2015; Silver et al., 2017), but also in real-world applications (Gregurić et al., 2020). Futhermore, RL approaches are being considered for some of humanities most pressing problems, such as the need to build sustainable food supply (Binas et al., 2019) and energy forecasting systems (Jeong and Kim, 2020), brought about through global climate change (Manabe and Wetherald, 1967; Hays et al., 1976; Hansen et al., 2012; Rolnick et al., 2019).

**Our system.** We develop our system specifically in the context of cooperative mutli-agent RL (OroojlooyJadid and Hajinezhad, 2019), where multiple agents jointly attempt to learn how to act in a partially observable environment by maximising a shared global reward. Our agents make use of model-based reinforcement learning (Langlois et al., 2019; Moerland et al., 2020). To learn an artificial language of the imagination, each individual agent in our system is given access to a recurrent world model capable of learning rich abstract representations of real and imagined future states. We combine this world model with an encoder function to encode world model rollouts as messages and use a recurrent differentiable message passing channel for communication. To show the benefits of our system, we develop a set of ablation tests and specialised experiments using novel as well as well-known multi-agent environments and compare the performance of our system to a set of strong model-free deep MARL baselines.

**Our findings and contributions.** We find that joint planning using learned communication through imagination can significantly improve MARL system performance when compared to a set of state-of-the-art baselines. We demonstrate this advantage of planning in a set of specialised environments specifically designed to test for the use of communication combined with imagined future prediction.

Our present work is not at scale and we only consider situations containing two agents. However, to the best of our knowledge, this is the first demonstration of a model-based deep MARL system that combines world models with differentiable communication for joint planning, able to solve tasks successfully, where state-of-the-art model-free deep MARL methods fail. We see this work as a preliminary step towards building larger-scale joint planning systems using model-based deep multi-agent RL.

## 2 BACKGROUND AND RELATED WORK

Reinforcement learning is concerned with optimal sequential decision making within a particular environment. In single agent RL, the problem is modeled as a Markov decision process (MDP) defined by the following tuple $(\mathcal{S}, \mathcal{A}, r, p, \rho_0, \gamma)$ (Andreae, 1969; Watkins, 1989). At time step $t$, in a state $s^t$, which is a member of the state space $\mathcal{S}$, the agent can select an action $a^t$ from a set of actions $\mathcal{A}$. The environment state transition function $p(s^{t+1}|s^t, a^t)$ provides a distribution over next states $s^{t+1}$ and a reward function $r(s^t, a^t, s^{t+1})$ returns a scalar reward, given the current state, action and next state. The initial state distribution is given by $\rho_0$, with $s^0 \sim \rho_0$, and $\gamma \in (0, 1]$ is a discount factor controlling the influence of future reward. The goal of RL is to find an optimal policy $\pi^*$, where the policy is a mapping from states to a distribution over actions, that maximises long-term discounted future reward such that $\pi^* = \text{argmax}_\pi \mathbb{E}[\sum_{t=0}^{\infty} \gamma^t r(s^t, a^t, s^{t+1})]$. If the environment state is partially observed by the agent, an observation function $o(s^t)$ is assumed and the agent has access only to the observation $o^t = o(s^t)$ at each time step, with the full observation space defined as $\mathcal{O} = \{o(s)|s \in \mathcal{S}\}$. In this work, we focus only on the case of partial observability.

**Deep RL.** Popular algorithms for solving the RL problem include value-based methods such as $Q$-learning (Watkins and Dayan, 1992) and policy gradient methods such as the REINFORCE algorithm (Williams, 1992). Q-learning learns a value function $Q(s, a)$ for state-action pairs and obtains a policy by selecting actions according to these learned values using a specific action selector, e.g. $\epsilon$-greedy (Watkins, 1989) or UCB (Auer et al., 2002). In contrast, policy gradient methods learn a parameterised policy $\pi_\theta$, with parameters $\theta$, directly by following a performance gradient signal with respect to $\theta$. The above approaches are combined in actor-critic methods (Sutton et al., 2000), where

the actor refers to the policy being learned and the critic to the value function. In deep RL, the policy and value functions use deep neural networks as high-capacity function approximators capable of learning distributed abstract representations from raw input signals that are useful for downstream decision making. Recent state-of-the-art deep RL methods include Deep Q-Networks (DQN) (Mnih et al., 2013) and related variants (Hessel et al., 2017), as well as advanced actor-critic methods such as PPO (Schulman et al., 2017) and SAC (Haarnoja et al., 2018). See (Arulkumaran et al., 2017; Li, 2017) for an in-depth review of deep RL.

**Model-based RL.** In RL, the environment transition function $p$ is typically unknown. As a result, so-called model-free RL methods, such as DQN and PPO, rely solely on data gathered from the environment, i.e. *real* experience, to learn an optimal policy. However, if given access to a transition function, an agent can generate useful simulated, or *imagined* experience, and use it to plan. Therefore, in model-based RL, a model $\hat{p}_\phi(o^{t+1}|o^t, a^t)$ with parameters $\phi$ is learned using stored transitions gathered from either a random, heuristic or learned policy to simulate transitions from the true (unknown) transition function $p$. The model can then be used for model-based planning, which can either happen in the background, or at decision-time. We briefly highlight the differences between these two types of planning and discuss work related to each and how this relates to our own work.

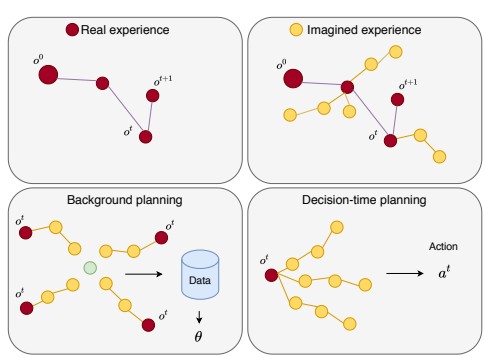

Figure 1: *Experience and planning in model-based reinforcement learning*

– *Background planning.* In background planning, the model is primarily used to generate additional experience and assist learning, i.e. for updating the parameters of the policy and/or value functions. An early version of this approach is DYNA-$Q$ (Sutton, 1990) which uses the additional experience to help learn a value function. However, the usefulness of a model degrades over long time horizons as model rollout error starts to compound (Gu et al., 2016). This has lead to different approaches that either use fixed depth rollouts based on model uncertainty (Feinberg et al., 2018), dynamic rollout schedules (Buckman et al., 2018), or short rollouts starting from intermediate states sampled from a buffer (Janner et al., 2019). A promising alternative approach is to update gradients directly via imagined rollouts in a lower-dimensional latent space (Hafner et al., 2019; 2020; Byravan et al., 2020).

– *Decision-time planning.* In decision-time planning, the model is used to generate imagined rollouts from a given state for the purpose of selecting the optimal action or sequence of actions. Decision-time planning methods for discrete action spaces often rely on search methods such as Monte Carlo tree search (MCTS) (Coulom, 2006) and have been used successfully in several works (Silver et al., 2017; Anthony et al., 2017; Schrittwieser et al., 2019). In continuous action spaces, methods include trajectory optimisation approaches using trajectory sampling (Todorov and Li, 2005; Theodorou et al., 2010; Nagabandi et al., 2018; Chua et al., 2018) or collocation (Posa et al., 2014) (optimising reward while forcing the model's predictions to be close to already visited states).

The model in our system is utilised for decision-time planning and follows the approach of Ha and Schmidhuber (2018), who used recurrent neural world models as a way to give agent's the ability to *learn how to think* (Schmidhuber, 2015). Specifically, we make use of a recurrent world model that takes the form of a mixture density network LSTM (MDN-LSTM), as used in (Ha and Eck, 2017). The model is therefore a form of a recurrent Gaussian mixture model and allows us to sample probabilistic predictions of imagined next states.

An illustration of the core features of model-based RL and the different types of planning is given in Figure 1. Also see (Janner, 2019) and (Mordatch and Hamrick, 2020) for useful overviews.

**Multi-agent RL (MARL).** In the multi-agent case with $N$ agents, we use the formalism of partially observable Markov games (Littman, 1994), defined as the tuple given above for the single agent case, but with observation and action spaces given by the following cartesian products: $\mathcal{O} = \prod_{i=1}^{N} \mathcal{O}_i \subseteq \mathcal{S}$

and $\mathcal{A} = \prod_{i=1}^{N} \mathcal{A}_i$, for agents $i = 1, ..., N$. The goal in this setting is to find an optimal joint policy $\pi^*(a_1, ..., a_N | o_1, ..., o_N)$ that maximises a shared long-term discounted future reward for all agent as $\pi^* = \operatorname{argmax}_\pi \mathbb{E}[\sum_{i=1}^{N} \sum_{t=0}^{\infty} \gamma^t r(o_i^t, a_i^t, o_i^{t+1})]$.

Early work in MARL simply trained multiple independent $Q$-learning algorithms (Tan, 1993), which has since been extended to include deep neural networks, or more specifically independent DQNs (Tampuu et al., 2017). However, from the perspective of an individual agent, these approaches treat all other learning agents as part of the environment, resulting in the optimal policy distribution to become non-stationary. Furthermore, if the environment is only partially observable, the learning task can become even more difficult, where agents may struggle with credit assignment due to spurious rewards received from unobserved actions of other agents (Claus and Boutilier, 1998).

To mitigate the issue of non-stationarity, MARL systems are often designed within the paradigm of *centralised training with decentralised execution* (CTDE) (Oliehoek et al., 2008; Lowe et al., 2017; Foerster et al., 2017). In CTDE, a centralised value function, or critic, is used during training, which conditions on the global state and joint actions from all the agents to make the learning problem stationary, but is then later removed once the individual agent's policies have been learned, making it possible to use each policy independently during system execution. However, individual agent policies extracted in this way may still perform poorly because training is not specifically aligned with the goal of performing well under decentralised execution. Therefore, state-of-the-art value-based MARL approaches such as Q-mix (Rashid et al., 2018) and QTRAN (Son et al., 2019) make use of value function decomposition strategies (Sunehag et al., 2017) to more closely resemble decentralised training, where each agent is a recurrent DQN (Hausknecht and Stone, 2015) that has memory to also deal with partial observability. Another clear way to help with the issue of partial observability is for agents to be able to communicate.

Learned multi-agent communication has been a key innovation in helping MARL systems to scale to more complex environments and solve more challenging tasks (Foerster et al., 2016; Sukhbaatar et al., 2016; Singh et al., 2018; Chu et al., 2020). To facilitate communication in our work, we formally extend the Markov game $\mathcal{M}$ by having agents connected to each other via communication channels according to a pre-defined neighbourhood graph $\mathcal{G}(\mathcal{V}, \mathcal{E})$. The graph $\mathcal{G}$ is defined by a set of nodes (vertices) $\mathcal{V}$ along with a set of edge connections $\mathcal{E} = \{(i,j)|i,j \in \mathcal{V}, i \neq j\}$, where each agent is a node in the graph, locally connected to other agent nodes. We define the connected neighbourhood surrounding agent $i$ as $\mathcal{N}_i = \{j \in \mathcal{V}|(i,j) \in \mathcal{E}\}$. This networked Markov game $\mathcal{M}_\mathcal{G}$ is then defined by the following tuple $(\mathcal{G}, \mathcal{S}, \mathcal{A}, r, p, \rho_0, \gamma)$. Our communication channels are recurrent and end-to-end differentiable allowing for agent-to-agent communication protocols to be learned during training. Unlike work studying the emergence of language through communication in MARL, e.g. (Lazaridou et al., 2016; Mordatch and Abbeel, 2017; Kajić et al., 2020) our work is more focused on communication through imagination as a useful system design for task solving, as apposed to uncovering new insights into emergent phenomena related to the human imagination.

**Model-based MARL.** To the best of our knowledge, the literature on model-based deep MARL is quite sparse and very little work has been done in this area. A notable exception is the recent work by Krupnik et al. (2020) on multi-agent model-based latent space trajectory optimisation. Here a multi-step generative model, specifically a temporal segment model (Mishra et al., 2017), is used to generate rollouts in a disentangled latent space and optimisation is performed directly over agent latent variables. Our work is the first we are aware of in the area of model-based deep MARL that combines communication with decision-time planning using recurrent neural world models.

## 3 METHOD

In this section, we provide the full details of our approach to model-based deep MARL and outline our system architecture, which we refer to as MACI: **M**ulti-**A**gent **C**ommunication through **I**magination. We explain the details of the system by way of a walk-through from the perspective of a single agent $i$, from time step $t$ to $t + 1$. At time step $t$, the agent receives the observation $o_i^t$ and initiates an imagined rollout of possible future observations and predicted rewards.

**Rollout.** For $k = 1, ..., K$ rollout steps, the agent produces an action:

$$a_i^k = \text{AgentController}_{\text{MLP}}(o_i^k, h_i^{k-1}, m_i^{I,c-1}), \tag{1}$$

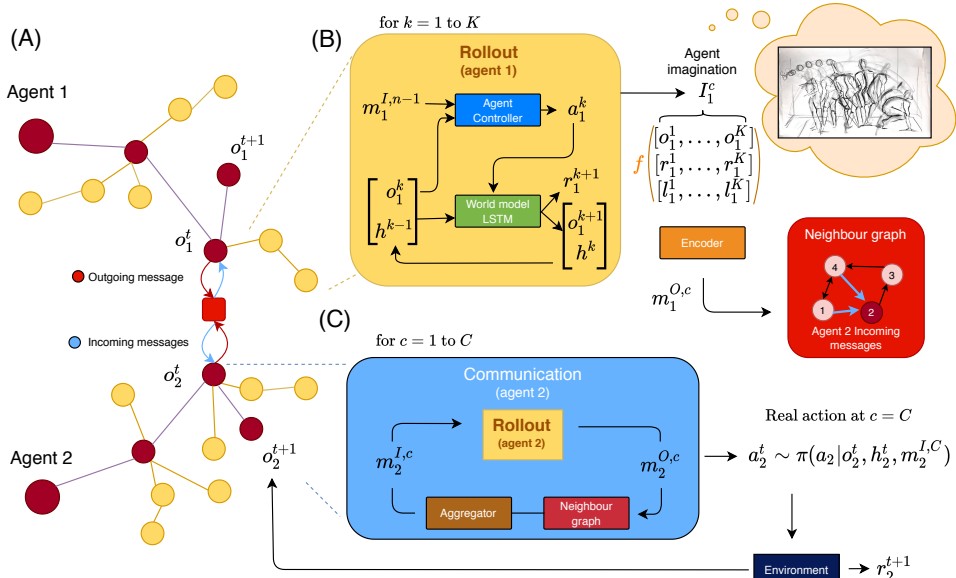

Figure 2: *MACI: Multi-Agent Communication through Imagination* – system architecture. **(A)** Agent trajectories, real and imagined. **(B)** Rollout for agent 1. **(C)** Communication for agent 2.

using a neural network (MLP) controller (we provide specifics, e.g. layer sizes and number of layers in the appendix), where $o_i^k$ is an imagined observation, $h_i^{k-1}$ is the world model hidden state, and $m_i^{I,c-1}$ is an aggregated incoming message to agent $i$, from connected agents in agent $i$'s neighbourhood, i.e. agents $j \in \mathcal{N}_i$. In turn, an imagined rollout is produced by the world model given an observation $o_i^k$ and an action $a_i^k$:

$$o_i^{k+1}, r_i^{k+1}, h_i^k = \text{WorldModel}_{\text{MDN-LSTM}}(o_i^k, h_i^{k-1}, a_i^k), \tag{2}$$

where the world model output includes the imagined next observation $o_i^{k+1}$, reward $r_i^{k+1}$ and an updated hidden state $h_i^k$. To initialise the rollout procedure, we set $o^{k=1} = o^t$, $h^{(k-1)=0} = h^{t-1}$ and $m_i^{I,(c-1)=0} = m_i^{I,t-1}$. The final output of the rollout after $K$ steps is the agent's imagination of the future summarised as follows: $I_i^c = \text{CONCAT}([o_i^1, ..., o_i^K], [r_i^1, ..., r_i^K], [l_i^1, ..., l_i^K])$, where $l_i^k$ are the logits associated with $a_i^k$ to maintain differentiability and we concatenate along a depth dimension to maintain a sequence of length $K$. Once the agent's rollout is complete, the agent starts to communicate its imagination to its neighbours.

**Communication.** For $c = 1, ..., C$ communication rounds the agent encodes it's imagination into a summarised abstract representation to serve as an outgoing message:

$$m_i^{O,c} = \text{ENCODER}_{\text{1D-CNN}}(I_i^c), \tag{3}$$

and sends this message to its connected neighbours. To encode the imagined sequence, we use a 1D convolutional neural network (CNN). In return, the agent receives all outgoing messages $m_j^{O,c}$ from its neighbours $j \in \mathcal{N}_i$, which is then turned into an incoming message using a simple aggregation function (e.g. concatenation, average or summation):

$$m_i^{I,c} = \text{AGGREGATOR}(\{m_j^{O,c} | j \in \mathcal{N}_i\}). \tag{4}$$

Note that for the next round of communication to begin, another rollout inner loop of $K$ steps must first take place to produce an updated imagination $I_i^{c+1}$. After $C$ communication rounds the agent takes a real action $a_i^t = \text{AgentController}_{\text{MLP}}(o_i^t, h_i^t, m_i^{I,C})$, conditioned on the final message $m_i^{I,C}$ and receives from the environment the next real observation $o^{t+1}$ and reward $r^{t+1}$. Finally, for our agents, we employ weight sharing and make use of a single world model shared between all agents. An illustration of the MACI system with two agents is provided in Figure 2.

**Training.**

**Block 1:** MACI – Methods

AgentController: $f$
WorldModel: $g$
Encoder: $z$
Aggregator: $h$

**Function** playEpisode (*f, g, z, e=None*):
$\quad o_1^0, ..., o_{|\mathcal{V}|}^0 \sim p_{env}(o_i, e|i \in \mathcal{V})$
$\quad$**for** $t = 1, ..., T$, *environment steps* **do**
$\quad\quad$ # Plan
$\quad\quad$**for** $c = 1, ..., C$, *communication steps* **do**
$\quad\quad\quad$ # Imagine
$\quad\quad\quad$**for** *agent* $i \in \mathcal{V}$ **do**
$\quad\quad\quad\quad$**for** $k = 1, ..., K$, *rollout steps* **do**
$\quad\quad\quad\quad\quad$ act:
$\quad\quad\quad\quad\quad$ $a_i^k = f(o_i^k, h_i^{k-1}, m_i^{I,c-1})$
$\quad\quad\quad\quad\quad$ imagine:
$\quad\quad\quad\quad\quad$ $o_i^{k+1}, r_i^{k+1}, h_i^k =$
$\quad\quad\quad\quad\quad\quad g(o_i^k, h_i^{k-1}, a_i^k)$
$\quad\quad\quad\quad$**end**
$\quad\quad\quad\quad$ consolidate:
$\quad\quad\quad\quad$ $I_i^c =$
$\quad\quad\quad\quad\quad$ CONCAT$([o_i^1, ..., o_i^K], [r_i^1, ..., r_i^K],$
$\quad\quad\quad\quad\quad\quad [l_i^1, ..., l_i^K])$
$\quad\quad\quad\quad$ encode outgoing message:
$\quad\quad\quad\quad$ $m_i^{O,c} = z(I_i^c)$
$\quad\quad\quad$**end**
$\quad\quad\quad$ # Communicate
$\quad\quad\quad$**for** *agent* $i \in \mathcal{V}$ **do**
$\quad\quad\quad\quad$ aggregate incoming messages:
$\quad\quad\quad\quad$ $m_i^{I,c} = h(\{m_j^{O,c}|\text{for } j \in \mathcal{N}_i\})$.
$\quad\quad\quad$**end**
$\quad\quad$**end**
$\quad\quad$ # Step
$\quad\quad$ act:
$\quad\quad$ $a_i^t = f(o_i^t, h_i^t, m_i^{I,C})$
$\quad\quad$ observe:
$\quad\quad$ $o_i^{t+1} \sim p_{env}(o_i, e|a_i^t)$
$\quad$**end**
$\quad$**return** $o^1...o^T, a^1...a^T, r^1...r^T$
**End Function**
# Loss functions
$\mathcal{L}_k^g(\theta) = (o^t - g_{o,\theta}(h^{t-1}))^2 + c(r^t - g_{r,\theta}(h^{t-1}))^2$
$\mathcal{L}_k^{f,c}(\theta) = \text{PPO\_loss}(\theta, A, \epsilon)$

---

**Block 2:** MACI-Training

Initialize $f$, $g$ and $z$;
**for** $e = 1, ..., E$, *training steps* **do**
$\quad$ Initialize experience buffer ($ex$)
$\quad$**for** $n = 1, ..., N$, *episode steps* **do**
$\quad\quad$ $ex$ += playEpisode(f, g, z);
$\quad$**end**
$\quad$ # Update world model parameters
$\quad$**for** $e$ *in* $ex$ **do**
$\quad\quad$ playEpisode(f, g, z, e);
$\quad\quad$ $\theta_{k+1}^g =$
$\quad\quad$ $\text{argmax}_\theta \frac{1}{\mathcal{B}T} \sum_{\tau \in \mathcal{B}} \sum_{t=0}^T \mathcal{L}_k^g(\theta^g)$
$\quad$**end**
$\quad$ # Update policy and encoder parameters
$\quad$**for** $e$ *in* $ex$ **do**
$\quad\quad$ playEpisode(f, g, z, e);
$\quad\quad$ $\theta_{k+1}^{f,z} =$
$\quad\quad$ $\text{argmax}_\theta \frac{1}{\mathcal{B}T} \sum_{\tau \in \mathcal{B}} \sum_{t=0}^T \mathcal{L}_k^{f,z}(\theta^{f,z})$
$\quad$**end**
**end**

---

**Block 3:** Python code outline

```python
def play_episode():
    obs = env.reset()
    states = zeros()  # {id: (world_state, comm_state,
      message), ...}

    for time_step = 1,..., T if not done:
        actions, states = next_actions(obs, states)
        obs, reward, done = env.step(actions)

def next_actions(obs, states):
    action_values, states = action_values(obs, states)
    actions = {id: argmax(q) for id, q in action_values
      }

    # keep world model in sync with env
    states = update_world_models(obs, actions, states)

    return actions, states

def action_values(obs, states):

    for step = 1,..., comm_rounds:
        states = update_comm_nets(obs,
      aggregate_messages(obs, states))

    return {
        id: agent.controller_net(ob, state)
        for id, agent, ob, state in zip(agents, obs,
      states)
    }, states

def aggregate_messages(obs, states):
    message_from = {
        id: agent.encode_plan(ob, state)
        for id, agent, ob, state in zip(agents, obs,
      states)
    }

    message_to = {
        id: mean(message_from[other] for other in adj)
        for id, adj in comm_graph
    }

    return {
        id: state.update_message(message)
        for id, state, message in zip(states,
      message_to)
    }

def agent.encode_plan(current_ob, state):
    obs, action_values, rewards = [], [], []

    for step in 1,..., rollout_steps:
        obs.append(current_ob)
        action_values.append(self.controller_net(obs
      [-1], state))

        current_ob, reward, hidden = self.world_model(
            obs[-1], argmax(action_values[-1]), state.
      world_state,
        )

        state = state.update_world(hidden)
        rewards.append(reward)

    return self.encoder(obs, action_values, rewards)
```

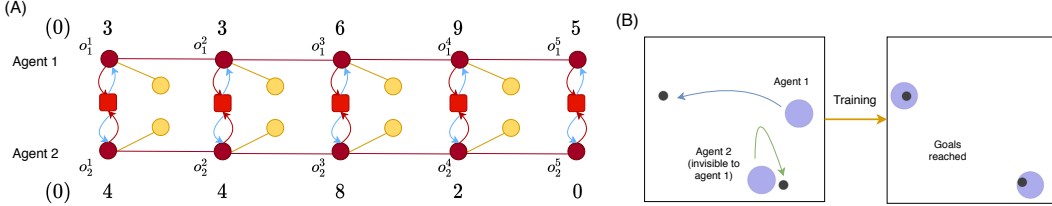

Figure 3: *Specialised experimental environments.* **(A)** Digit game. **(B)** Invisible spread.

## 4 EXPERIMENTS

We test the feasibility of our system on a set of specialised environments. Each environment is conceptually simple. However, both environments still prove to be a formidable challenge, even for state-of-the-art MARL approaches, due their extreme partial observability and the need for agents to be able to think ahead *and* communicate to solve the task. Our first environment, *Digit game*, is inspired by a similar challenge in (Foerster et al., 2016) and our second environment, *invisible spread*, is based off of the popular multi-agent particle environment (Mordatch and Abbeel, 2017; Lowe et al., 2017). Similar to previous work on model-based MARL (Krupnik et al., 2020), we only consider the case of two agents. In all our experiments, we compare our system against basic baselines namely, independent PPO learners (IPPO) (a surprisingly strong baseline) and independent $Q$-learners (IQL), as well as state-of-the-art systems, Q-mix and QTRAN. Futhermore, in each experiment we use short two-step rollouts to guard against compounding model error.

**Digit game.** At each time step, each agent receives a random one-hot encoded digit (0-9), where the environment transition dynamics obeys the following rule: $o^{t+1} = (o^t + o^{t-1}) \mod 10$, with $o^0 = 0$. The challenge is for each agent to produce as action the digit the other agent is likely to receive at the next time step. If the agent action is correct, the agent gets a reward of $1$, if the action is incorrect, it gets $0$. The digit game therefore specifically tests if agents can predict future observations based on present and past experience (i.e. use their imagination), as well as learn an effective communication protocol (as the other agent's observation can only be determined through communication). Figure 3 (A) shows an example of the digit game. In this example, if agent $1$ selects as action the digit $8$ at time step $t = 2$, i.e. $a_1^2 = 8$, it will receive a reward of $1$.

– *Results.* We provide our experimental results on the digit game environment in Figure 4 (A). Learning curves are mean rewards averaged over 5 runs and include shaded standard deviation bounds. Our system, MACI, is shown to have a clear advantage, significantly outperforming all the other baselines. Due to the fact that we make use of a world model to perform rollouts that cost compute time, we also show (in the inset plot) the wall-clock time for each system over real environment steps. Although MACI scales less well than the baselines in this specific instance, we note that in more complex scenarios, real environment steps may prove more expensive than imagined rollouts and the sample efficiency of decision-time planning could outweigh the extra model compute time.

**Invisible spread.** The goal in this environment is for each agent to occupy a separate landmark starting from a random location. Agents receive a shared reward based on how close both agents are to their respective landmarks. The observation space consists of values for the agent's position and velocity as well as the positions of the landmarks. To make the task more difficult, we make each agent invisible to the other. Therefore, agents must coordinate through communication and use their world models to show their intent (in terms of the landmark they are currently planning to occupy). Figure 3 (B) shows an example of the invisible spread environment, where the agents are represented as large purple circles and the landmarks are shown as small black circles.

– *Results.* We provide our results on the invisible spread environment in Figure 4 (B). Learning curves are again mean rewards averaged over 5 runs and include shaded standard deviation bounds. MACI is shown to again have a clear advantage, outperforming all the other baselines. Interestingly, in this environment, MACI scales well in terms of compute time and is shown to perform close to the most efficient baseline, IPPO.

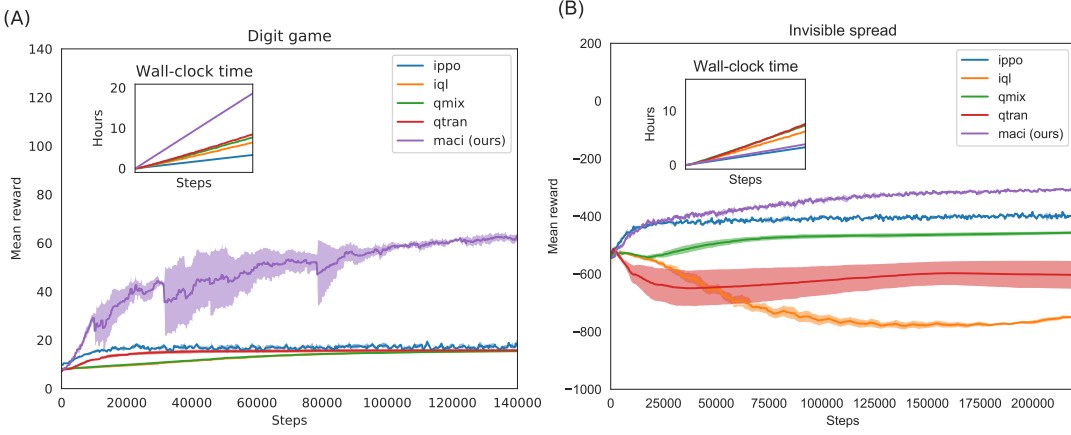

Figure 4: *Experimental results* (**A**) Digit game. (**B**) Invisible spread.

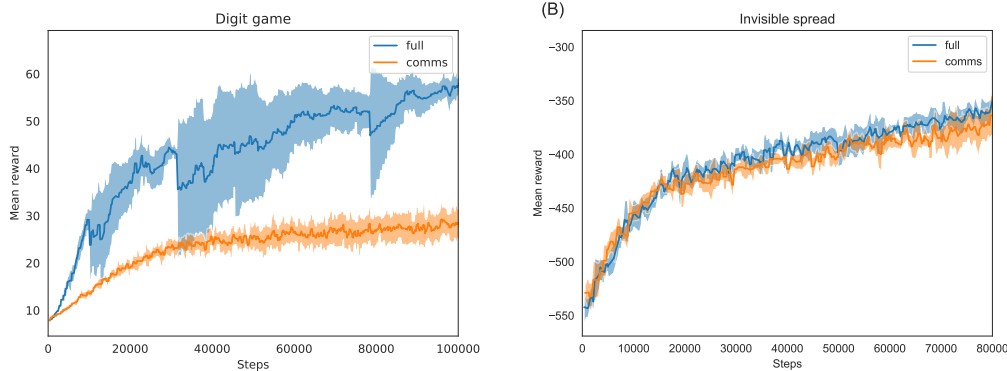

Figure 5: *Ablation study* (**A**) Digit game. (**B**) Invisible spread.

**Ablation study.** To disentangle the roles of communication and imagination we perform an ablation study on both environments, digit game and invisible spread. In each case, we train the MACI system with and without using a world model. We perform 5 independent runs for each case, showing the average curves with standard deviations bounds. The results of this study is shown in Figure 5. For the digit game, shown in panel (A), the world model is a crucial component determining final performance. This is to be expected given the strong requirement for future prediction. In the invisible spread environment, the benefit is less significant and the system seems to rely more heavily on communication of past and present information.

# 5 CONCLUSION

Our ability to imagine plays an integral role in our intelligence. Inspired by the idea of language as the *systematic instruction of imagination*, we developed a system for multi-agent communication through imagination (MACI) in the context of model-based deep mutli-agent RL. In our system, each agent has access to a recurrent world model for decision-time planning and uses a differentiable message passing channel for learned communication.

In our experiments on two specialised environments, digit game and invisible spread, we showed that using learned communication through imagination can significantly improve MARL system performance when compared to state-of-the-art baselines. Although our environments are conceptually simple, both environments still proved to be a formidable challenge, even for state-of-the-art methods. Furthermore, the sample efficiency of decision-time planning was shown to outweigh the extra model compute time in the invisible spread environment.

Our work demonstrates the feasibility of a model-based deep MARL system that combines world models with differentiable communication for joint planning. Specifically, it highlights the potential benefits of decision-time planning in a multi-agent setting as a means to improve agent cooperation.

An interesting future line of work could explore the combination of background and decision-time planning in the multi-agent setting. In addition, many interesting innovations from the single agent model-based RL literature could potentially find fruitful application in MARL. However, scaling MARL to larger numbers of agents remains a difficult task, and even more so for model-based methods. We see this work as a first step towards building larger-scale joint planning systems, which we plan to undertake in future work.

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

# 6 APPENDIX

---

**Block 4:** MACI – Settings

---

**Environment**:
- obs_dim, $D_{obs}$
- action_dim, $D_{act}$

**Hyperparameters**:
- message_dim, $D_m$ = 16
- agent_controller_hidden_dim, $D_a = 16$
- agent_controller_hidden_layers, $H_a = 1$
- world_model_hidden_dim, $D_{wm} = 16$
- world_model_hidden_layers, $H_{wm} = 1$
- rollout steps, $K$
- communication steps, $C$

**Architectures**:
- Agent controller, $f(\cdot)$:
    *Type*: Feedforward MLP
    *Input dim*: $D_m + D_{obs} + D_{wm}$
    *Hidden dim*: $D_a$
    *Hidden layers*: $H_a$
    *Output dim*: $D_{act}$
- World model, $g(\cdot)$:
    *Type*: LSTM
    *Input dim*: $D_{obs} + D_{act}$
    *Hidden dim*: $D_{wm}$
    *Hidden layers*: $H_{wm}$
    *Output dim*: $D_{obs} + 1$ (reward)
- Encoder, $z(\cdot)$:
    *Type*: 1D convnet
    *Input width*: $D_{obs} + D_{act} + 1$ (reward)
    *Input length*: $K$
    *Output dim*: $D_m$
- Aggregator, $h(\cdot)$:
    *Type*: Concatenation
- Neighbour graph:
    *Type*: Fully connected

---

