# OpenReview forum: "Learning to communicate through imagination with model-based deep multi-agent reinforcement learning"
_ICLR.cc/2021/Conference — Reject_

### Official Review · AnonReviewer3 · 2020-10-26
**The presented algorithm is interesting, but the paper needs reframing and improved experimental method**

**Rating:** 3
**Confidence:** 3

**Review:**


This paper claims to present an algorithm which enables a population of (two) agents
to learn to communicate and coordinate to solve a task, and thus positions itself
in the field of multi-agent Deep RL. After a long but rather vague and unspecific introduction
and related work (see below), it describes the algorithm, then presents experiments where
the introduced algorithm is compared with model-free MARL baselines.

While the algorithm presented is interesting and has potentially some novelties compared to
the state-of-the-art (e.g. differentiability of message passing in model-based MARL), it has
also a number of weaknesses:

1) Globally, I had a lot of difficulty understanding clearly what are the aims of this paper:
What are the problems it aims to solve? What are the scientific questions adressed?
Neither the abstract nor the text provide sharp explanations of these aims.

2) The paper uses very loaded but undefined vocabulary like "imagination", "language" and "communication".
While in general I think it can be sometimes useful to use concepts and terms from human cognitive sciences
to describe AI systems, in this particular case I found it very far fetched to speak of "imagination" and "language",
even "communication". It seems in practice authors might simply mean something like "prediction of future states"
when they use the term "imagination". "Language" and "communication" are also far-fetched because in cognitive
science and linguistics it refers to systems that enable different individuals, with different world views, to
communicate an intent to each other. Here, the "agents" share the same world model, so they are not really
different individuals with their own world representations, and their communication is rather like
message passing in GNNs, which is pretty far from "language" or "human-like communication".

3) It is not even clear whether it is meaningful to call the presented system as "multi-agent", since in addition
to a centralized shared reward, there is also a shared world model. To me, the system looks rather like an RL
system that controls a multi-component body with local controllers that synchronize through message passing,
quite similary to graph neural network controllers (also including message passing) used for e.g. in Pathak et al. 2019.
A discussion of the similarities and differences with work such as Pathak et al. is needed.

4) the authors are right to say that there is little research on model-based MARL, and cite one exception:
Krupnik et al. However, it is not justified why this closely related work is not included in the baselines,
or at least compared in discussion more thoroughly. Authors might also want to discuss another model-based MARL
paper: Zhang et al. 2020.

5) A large part of the related work section is not relevant to this paper, in particular about Deep RL and model-based RL,
which are much broader topics than the one addressed in this paper

6) The description of the method lacks sufficient technical details for reproducibility, in particular it lacks detailed
pseudo-code (some refs are said to be in an appendix, but I did not find an appendix), and no links to code is provided.
Furthermore, there is no sufficient information on how hyperparameters selection for baselines was made.

7) The two environments in the experiments are not sufficiently well motivated: why did you need to introduce them rather
than reuse existing test environments? E.g. which particular problems did you want to address that was not possible with
existing environments ?

8) Since the claimed topic of the paper is about the emergence of a "communication system", one would expect a detailed
analysis of the emergent communication code (currently only figure 5 gives a quite superficial qualitative analysis).

9) The quantitative comparison of algorithms is not made using a sufficiently strong statistical method (only 5 seeds,
no tests such as Welch t-tests)

For these reasons, while the particular algorithms studied is in itself interesting, I think the paper would need a major
conceptual reframing and a better experimental methodology and justification before publication.

References:

Pathak et al. (2019) Learning to Control Self-Assembling Morphologies: A Study of Generalization via Modularity
https://arxiv.org/pdf/1902.05546.pdf

Zhang, K., Kakade, S. M., Başar, T., & Yang, L. F. (2020). Model-based multi-agent rl in zero-sum markov games with near-optimal sample complexity. arXiv preprint arXiv:2007.07461.

---

### Official Review · AnonReviewer4 · 2020-10-26
**A neat idea that requires further investigation**

**Rating:** 4
**Confidence:** 5

**Review:**

Thank you very much for sharing these cool ideas. I enjoyed the clear writing and excellent related work sections, and I genuinely believe this paper presents interesting concepts that warrant further investigation. Unfortunately, in its current state, this manuscript is not ready for to be shared with the wider community at ICLR.

I will leave here a few suggestions for improvement and ideas on how to strengthen your argument. I sincerely hope you will find these useful as you continue your research on this topic.

The manuscript describes Multi-Agent Communication through Imagination (MACI). MACI is an imagination-inspired communication protocol that allows two sub-modules to exchange information about their non-overlapping observations.

The manuscript is well written and easy to follow, and the authors properly place their contributions in the context of existing ideas.

While the experiments presented are clear and the results are encouraging, I think the experimental section could benefit from additional experiments, here is why:

- The tasks presented here are extremely simple. I understand the need of didactic environments, but in a purely methods paper, the reader is left to wonder if this method scales to more complex environments, if it can work with more than two agents, and if it can handle non-cooperative settings. This is especially acute here, given that Fig. 6 suggests MACI only helps in 1 out of 2 environments, as the performance gains in Invisible Spread are obviously attributable to partial observability in the baselines.

- The tasks presented are purely cooperative, and the communication system is differentiable. This means that by setting WorldModel, Encoder and Aggregator to the identity function, one would recover exactly a single-agent architecture that has access to the combined observations and operates in the product of the actions spaces. The only difference might lie in how the the system is supervised (it is unclear from the manuscript how WorldModel is trained). This is similar to what is presented in Fig. 6 in the ablation study, but would add including a shared world-model to produce an "ideal" agent. How does this perform? The baselines provided are at an obvious disadvantage as the environments are partially observable. This performance ceiling would guide the reader in understanding how much of the gap is recovered by MACI.

Additional minor remarks:

- I cannot find in the methods section how WorldModel is trained. Could this be made clearer in the text?
- How accurate is WorldModel? How important is this accuracy? What happens if we replace our learned WorldModel module with an ideal oracle?
- There is a bunch of work in modeling MARL (see, e.g. Hierarchical Policy Models [Zheng 2016], VAIN [Hoshen 2017], NRI [Kipf 2018] and RFM [Tacchetti 2018]). In particular RFM introduces on-board imagination models that influence the decisions of each agent. It might be good to add these to your references.

Thank you again for sharing these cool ideas, I hope to see more of this soon and that you'll find some of this feedback useful.
All the best.

---

### Official Review · AnonReviewer2 · 2020-10-28
**The claim is not well-supported.**

**Rating:** 4
**Confidence:** 3

**Review:**

Summary:


This paper proposes to combine model-based and multi-agent reinforcement learning. The authors follow the typical recurrent neural world models setting to generate imagined rollouts for decision-time planning. To tackle the non-stationarity of a multi-agent environment, they build end-to-end differentiable communication channels between agents within a pre-defined neighborhood. The communication message is defined as abstract information encoded from the imagined rollout. Agents then make decisions based on the message they received and the output of recurrent neural world models. Empirical studies are performed to show the superiority of proposed methods over SOTA model-free MARL approaches. Results are shown in two simple environments, which are designed to require communication between agents to solve the task.


##########################################################################

pros:

+ The motivation of doing model-based MARL is very clear and challenging.

+ Overall, the paper is well written.

+ The ablation study on the roles of world models and communication channels is interesting.

##########################################################################

cons:


- Although the paper claims as a combination of model-based and multi-agent RL, my major concern is that the proposed model still deals with these two problems separately. In particular, the world model doesn't consider the dynamics of other agents, thus being an independent model only. The paper proposed to tackle the multi-agent part of the problem by building an explicit communication channel, which lacks enough novelty.

- I'm also concerned about the lack of rigorous experimentation to support the paper's claim.
The two proposed environments are extremely tailored for algorithms with explicit communication channels and are limited in the number of agents.
	- For the digit game, the non-stationarity is not quite clear when there are only two agents. I'd like to see what would happen if the agent number in the digit game increases.
	- For the invisible spread, the ablation study shows that the role of world models is not important. I'd like to see the performance of other baseline algorithms that use explicit communication channels, which is not compared and seems to work well as the paper reported. If so, I don't see why this experiment supports the claim of combining model-based and multi-agent RL.

##########################################################################

Post rebuttal

The author's response does not address my primary concern and I'd like to keep my original score.

---

### Official Review · AnonReviewer1 · 2020-10-28
**Insufficient evidence for an otherwise interesting take on MARL algorithms**

**Rating:** 3
**Confidence:** 5

**Review:**

The paper talks about developing a model-based method for cooperative multi-agent reinforcement learning. The proposed approach utilizes communication as a tool for mitigating the partial observability induced by the non-stationary task while also helping agents reason about other agents' behaviors. The authors present their motivation for using language as a medium in model-based RL stemming from early literature in psychology and linguistics.

The setup consists of decentralized agents each of which is equipped with a world model similar to Ha et al. 2018. Further, each agent also has a separate message input that is received from the other players. Each agent does a form of decision-time planning where it produces rollouts for K steps before taking a real action. The message is then the encoding produced by the concatenation of the observations, rewards, and the actions taken during the rollouts.

The approach is novel and one of the first works that combine model-based RL in a dec-POMDP. The paper does a good job of explaining prior work in related domains. The schematic diagram also depicts the setup in an efficient and standalone manner.

Still, I have some qualms related to the experimental setup that arguably makes the contribution of the proposed imagination framework inconclusive.

- In the digits game, the agents need to produce actions that represent the next observation of the other agent. The transition dynamics are defined in a way such that the next observation for an agent i is independent of the action taken at the current timestep. I find this formulation to be incoherent with the way MACI works. Specifically,
     a) The AgentController that produces the action doesn't need to depend on the current observation since it has no effect on the action.
    b) The WorldModel produces the next observations, next hidden states, and the rewards given the current observation, current action, and current hidden state. Similar to the above, the information about the current action is not needed to produce the next observation. Moreover, the rewards, in this case, are only tied to the action. So it would make sense to produce it along with the action in the AgentController with a recurrent network.
Overall I believe this game is not aligned with the objectives of MACI, although I would love to have the authors clarify this.

- There is no information about the objective functions used for optimization or any detail about the learning process without which it makes it hard to reproduce.

- The choice of baselines doesn't seem to be appropriate for the task. Since all the baseline methods used do not use explicit communication in their original forms, the comparison thus becomes unfair. I would like the authors to reference if the baselines were modified in a way to accommodate this. This is important specifically in the two tasks chosen since I believe just adding communication should yield sufficient improvement.

- The current approach is only applicable for a two-agent cooperative game narrowing down the scalability of the method. I believe the approach has the potential to extend to multiple agents either by having a confluence of messages or explicit grouping of agents.

- An important missing ablation experiment is comparing comm+world model with only world model. This is crucial since it will determine whether the performance gain is due to the abstract planning or the communication.

- The overall compute required is more than running a real-time experiment since the planning uses K-step rollouts. Some ablation of the choice of K would be interesting to look at especially in terms of wall time.

typo: Fig 6-A title

---

### Author Response · Authors · 2020-11-24
**Response to comments**

We would like to start by thanking every reviewer for their valuable feedback. We have taken note of everything that was said and made the necessary improvements to our system. As can be seen in our updated paper, in Figure 4 and 5, the MACI algorithm is more stable than before, while still outperforming all other algorithms we tested against. This is due to our system now training the world model, policy and encoder in an iterative fashion. Previously the world model was only trained once. We provide a more detailed algorithmic setup of our new MACI algorithm on page 6.

Some of the reviewers rightfully pointed out that our digit game environment was quite limiting as agent actions did not influence future observations. We, therefore, created a new grid world environment that is much closer to the original RL setting. This environment allows us to test communication and navigation, where every action can influence future observations of all agents. With our new algorithm, we are able to scale beyond our previous limitation of 2 agents. We see that 4 agents can effectively communicate in this grid world. We also see that a combination of communication and a world model is needed for good performance.

The reviewers also pointed out that our paper lacks details on network architectures used for this work. We, therefore, added an Appendix with these specifications. Unfortunately, due to our main author falling sick, we did not get to address all the envisioned improvements and write them up in time. We will, however, keep on improving the algorithm in the coming weeks and release our update results.

Thank you for your time and consideration.

---

### Decision · Program_Chairs · 2021-01-07
**Final Decision**

**Decision:**

Reject

**Comment:**

The authors present a model-based method for cooperative multi-agent reinforcement learning and propose to use communication of future predictions (as given by a learned world model) as a way to overcome partial observability.

Overall, all reviewers found this work to be of great interest and the combination of planning + communication novel. However, all reviewers pointed that the claims that the papers makes are not fully supported by the experimental framing of the paper pointing to several shortcomings around experimental design in general and better control of appropriate baselines. The authors have since clarified several aspects in their paper and also included a new RL environment.

However, as the paper still stands does not fully provide convincing evidence of their proposal, which is however very intriguing. I would like to echo though reviewers' suggestions that the authors work a bit more on the experimental design and I really hope this work will appear at a later venue.